# Neural Phrase-to-Phrase Machine Translation

## Abstract

We present Neural Phrase-to-Phrase Machine Translation (NP$^2$MT), a phrase-based translation model that uses a novel phrase-attention mechanism to discover relevant input (source) segments to generate output (target) phrases. We propose an efficient dynamic programming algorithm to marginalize over all possible segments at training time and use a greedy algorithm or beam search for decoding. We also show how to incorporate a memory module derived from an external phrase dictionary to NP$^2$MT to improve decoding. Experiment results demonstrate that NP$^2$MT outperforms the best neural phrase-based translation model (Huang et al., 2018) both in terms of model performance and speed, and is comparable to a state-of-the-art Transformer-based machine translation system (Vaswani et al., 2017).

## 1 Introduction

Segmental structures (i.e., phrasal information) are abstractions that have been shown to be useful in machine translation. State-of-the-art machine translation systems prior to the deep learning revolution were dominated by phrase-based models (Koehn et al., 2003; 2007; Lopez, 2008; Koehn, 2009). In contrast to word-based approaches, these methods consider explicit phrase structures in both source and target sentences and model their alignments.

Recent advances in sequence-to-sequence learning (Sutskever et al., 2014) and attention-based mechanisms (Bahdanau et al., 2014) have driven impressive progress on a new generation of machine translation models. They typically consist of three main components: an encoder that summarizes the source sentence into vectors, a decoder that generates the translation word by word, and a word-based attention module that is used by the decoder to capture relevant source information when decoding. Innovations in terms of model architectures (Vaswani et al., 2017; Gehring et al., 2017a) and optimization algorithms (Ba et al., 2016) push these models to become the de facto state-of-the-art machine translation systems. These models operate at the word-level, since discovering phrases for both the source and target sentences and incorporating this phrasal information into a neural sequence to sequence model is not straightforward (i.e., it requires marginalizing over all possible phrase alignments in a neural model).

Huang et al. (2018) show a first-step toward a neural phrase-based machine translation model by designing a model that is aware of phrase structures in the target sentence. They show that such an approach leads to better results compared to baseline sequence to sequence models on several language pairs.

In this paper, we present the first neural phrase-based machine translation model (NP$^2$MT) that considers phrasal information in both the source and target sentences. Our model uses segmental recurrent neural networks (Kong et al., 2015) to discover phrases on the source sentence and a novel phrase-attention model to align them to generated target phrases (§2.1). We design a dynamic programming algorithm to efficiently marginalize over all possible segments (phrases) in the source and target sentences in our model for training (§2.3) and use a greedy algorithm or beam search for decoding (§2.4). We also show how to augment this model with a memory module consisting of phrase-to-phrase translation (i.e., a phrase dictionary) in a setup when there are many unseen

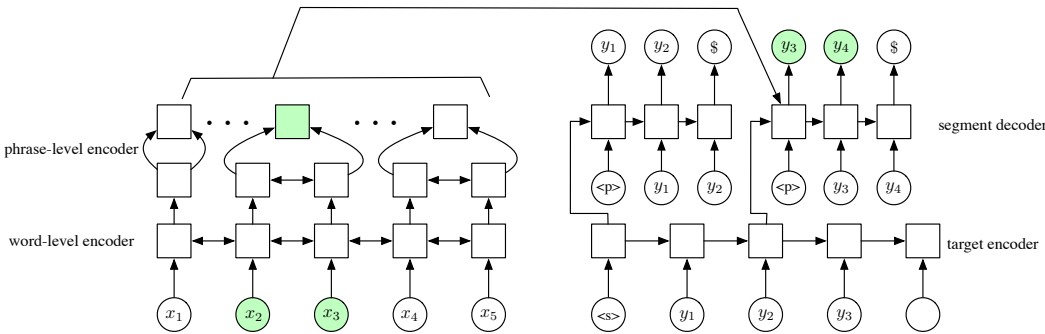

Figure 1: The architecture of NP$^2$MT model. The example shows how the target phrase $y_{3:4}$ is translated conditioning directly on the source phrase $x_{2:3}$, using the phrase-level attention. Note that for brevity, in the phrase-level encoder, we only show one possible segmentation of the source sentence. We use "$\cdots$" to indicate all the possible segments $x_{i:j}$ in Eq. 1.

words (e.g., proper nouns, cross-domain translation) to improve performance (§2.5). Experiments on benchmark datasets show that our model outperforms existing phrase-based machine translation models both in terms of performance and speed and is competitive with state-of-the-art word-based models (§3).

## 2   NP$^2$MT

Consider a source sentence $\boldsymbol{x}_{1:S} = \{x_1, x_2, \ldots, x_S\}$ and a target sentence $\boldsymbol{y}_{1:T} = \{y_1, y_2, \ldots, y_T\}$. Our model is based on the sequence-to-sequence model (Sutskever et al., 2014) with two main components: a source sentence encoder and a target sentence decoder. Figure 1 shows an overview of NP$^2$MT, we describe each component in details below.

### 2.1   ENCODER

We represent each word in $\boldsymbol{x}_{1:S}$ as a vector and use a bidirectional LSTM to obtain token representations $\mathbf{r}_1, \ldots, \mathbf{r}_S$ by concatenating the forward and backward hidden states.

We then obtain phrase representations using another bidirectional LSTM that takes token representations above as inputs based on Segmental RNNs (Kong et al., 2015). Specifically, in order to compute the representation of a phrase spanning from the $i$-th token to the $j$-th token, denoted by $\mathbf{s}_{i:j}$, we run a bidirectional LSTM with $\mathbf{r}_i, \ldots, \mathbf{r}_j$ as inputs. Denote the hidden states of the forward LSTM by $\overrightarrow{\mathbf{h}}$ and the hidden states of the backward LSTM by $\overleftarrow{\mathbf{h}}$. We have $\mathbf{s}_{i:j} = \text{concat}(\overrightarrow{\mathbf{h}}_j, \overleftarrow{\mathbf{h}}_i)$.

Given the maximum segment length $P$ (a hyperparameter of the model), we compute representations for all phrases:

$$\mathbf{s}_{i,j}, \forall (i,j) \in \{(i,j) : 1 \leq i \leq j \leq i + P - 1 \leq S\}$$

Since we use a bidirectional LSTM as the phrase encoder and all phrases are contiguous segments, we can efficiently compute these representations in $\mathcal{O}(SP)$.

### 2.2   DECODER

Let $\boldsymbol{z} \in \mathcal{Z}_{\boldsymbol{y}_{1:T}}$ be a valid of segmentation of $\boldsymbol{y}_{1:T} = \{y_1, y_2, \ldots, y_T\}$. Denote each segment in $\boldsymbol{z}$ by $z_k$ and the number of segments $|z_k|$.

The generative process of the target sentence is:

- For each segment $k = 1, \ldots, |z_k|$:
  - Sample a word $y_i^k$ until we get the end of segment symbol (i.e., $y_i^k = \$$) or the end of sentence symbol in the last segment (i.e., $y_i^k = \$\$$).

Our decoder consists of two main components:

- A unidirectional (forward) LSTM to compute a contextual representation $\mathbf{g}_{\boldsymbol{z}_{<z_k}}$ of previously generated segments $\boldsymbol{z}_{<k}$. In training time, $\mathbf{g}_{\boldsymbol{z}_{<k}}$ can be obtained simply by running the LSTM on all target words from previous segments token by token.
- A segment decoder that generates words in the current segment. The segment decoder is an attention-based model over all possible source phrases $\mathbf{s}_{i,j}$, $\mathbf{g}_{\boldsymbol{z}_{<z_k}}$, and previously generated words in the current segment $\boldsymbol{y}_{<t}^{z_k}$.

We compute the phrase-attention score $\mathbf{a}_{i,j}^{z_k}$ once for every segment $z_k$, using $\mathbf{g}_{\boldsymbol{z}_{<z_k}}$ to query all possible source phrase representations $\mathbf{s}_{i,j}$:

$$\mathbf{a}_{i,j}^{z_k} = \text{attn}(\mathbf{s}_{i,j}, \mathbf{g}_{\boldsymbol{z}_{<z_k}}) \tag{1}$$

The phrase-attention score $\mathbf{a}_{i,j}^{z_k}$ gives a soft alignment[1] between the target phrase $z_k$ and the source phrases $\mathbf{s}_{i,j}$. The attention value is then computed as the sum over $\mathbf{s}_{i,j}$ weighted by the attention score $\mathbf{a}_{i,j}^{z_k}$. We use it as the initial state of the segment decoder.

The probability of generating the target sentence is computed as:

$$p(\boldsymbol{y}_{1:T} \mid \boldsymbol{x}_{1:S}) = \sum_{\boldsymbol{z} \in \mathcal{Z}_{\boldsymbol{y}_{1:T}}} \prod_{t \in z_k} p(y_t^{z_k} \mid \mathbf{s}_{i,j}, \mathbf{g}_{\boldsymbol{z}_{<z_k}}, \boldsymbol{y}_{<t}^{z_k})$$

## 2.3 TRAINING

For a given pair of source and target sentence, the objective function that we would like to maximize in Eq. 2 is intractable due to the summation over $\mathcal{Z}_{\boldsymbol{y}_{1:T}}$ (i.e., all possible target segmentations). Here, we show a dynamic programming algorithm to compute it efficiently.

Denote the probability of generating all valid segmentations ending in the $n$-th target words by $\alpha(n)$. The probability of generating the entire sentence is therefore $\alpha(T)$.

We have the following recursive function:

$$\alpha(n) = \sum_{m<n} \alpha(m) \prod_{t \in z^{(m,n)}} p(y_t^{z^{(m,n)}} \mid \mathbf{s}_{i,j}, \mathbf{g}_{\boldsymbol{z}_{<z^{(m,n)}}}, \boldsymbol{y}_{<t}^{z^{(m,n)}}),$$

where $\alpha(0) = 1$, and $z^{(m,n)}$ denotes a segment from $m$ to $n$. The computational complexity of this dynamic programming algorithm is $\mathcal{O}(TP)$, where $P$ is the maximum (target) segment length.

## 2.4 DECODING

For inference at test time (i.e. decoding), we cannot use dynamic programming in §2.3 since the target sentence is unknown. We either use a greedy algorithm or beam search for decoding.

**Greedy algorithm** For the greedy method, we follow the generative process in §2.2 and greedily pick the word $y_i^k$, until we reach the end of sentence symbol (i.e. $y_i^k = \$\$$).

**Beam search** Our beam search algorithm is different from the standard left-to-right beam search algorithm used in vanilla sequence-to-sequence models since the probability for the next word is conditioned not only on the local context with in the segment, but also on all previously generated segments. The main idea of our algorithm is to keep track of both during the beam search process. We show our Algorithm in Appendix A.

We maintain a set $\mathcal{Y}_o$ to store the incomplete (open) sentences during the search. An open sentence is represented as a tuple consisting of three elements: previously generated segments, previously

---

[1]Enforcing a hard attention similar as in (Raffel et al., 2017) may be beneficial in the context of phrase-level attention. We leave that for future work.

generated words in the current segment, and the probability of generating the open sentence so far. This tuple contains sufficient information to compute the probability distribution of the next output word. We update each element (each open sentence) in $\mathcal{Y}_o$ after we generate a new word, so the open sentences in $\mathcal{Y}_o$ will always have the same length.

## 2.5 DECODING WITH A MEMORY MODULE

In language modeling, augmenting a recurrent neural network with a memory module at evaluation time has been shown to improve performance (Grave et al., 2016; Krause et al., 2017; Grave et al., 2017), especially for rare word predictions and in an open vocabulary setup. We explore the possibility of augmenting a neural machine translation model with a persistent memory module derived from external sources (i.e., a phrase-to-phrase dictionary) to improve decoding.

In this setup, we first build a phrase-to-phrase dictionary using a phrase extractor from Moses (Koehn et al., 2007). During decoding, at each timestep, we compute attention scores forr all possible source segments $\mathbf{a}_{i,j}^{z_k}$. We then choose the source phrase with the highest score to be translated. Given the attended source phrase, we decide whether to use a dictionary to translate the phrase or not. We perform a memory lookup and check whether the phrase is in the dictionary. If it is, we directly translate using the dictionary; otherwise, we proceed as in the greedy decoding case described above. When there are multiple translations for a phrase in the dictionary, we use the model to score all candidates, and then choose the best one as follows:

$$y^{z_k} \leftarrow \underset{d^{z_k} \in \mathbf{d}^{z_k}}{\arg\max} \prod_{t \in z_k} p(d_t^{z_k} \mid \mathbf{s}_{m:n}, \mathbf{g}_{\mathbf{z}_{<z_k}}, \boldsymbol{d}_{<t}^{z_k}) \quad (2)$$

We summarize our memory-augmented decoding method in Algorithm 1 and illustrate it in Figure 2.

We note that word-based models can also benefit from the augmented memory module. However, it is more straightforward to use the phrase-level information in NP$^2$MT. The word-based models only lookup phrases with source with length 1 using attention weights. Experiments show that when compared to models not augmented with memory modules, NP$^2$MT improves significantly when there is a larger amount of OOV (See Appendix Table 6).

In addition, it is also possible to augment beam search decoding with a memory module. In this case, we perform a lookup and we add the translated phrase to $\mathcal{Y}_o$ after the beam reaches the length of the translated phrase. We present the greedy case for simplicity. It should be straight-forward for the readers to extend the algorithm into beam search cases by combining algorithms 1 and 2.

We leave exploration of beam search and memory to future work.

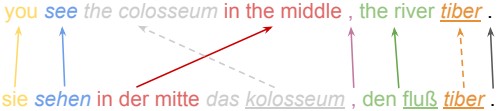

Figure 2: The decoding process of NP$^2$MT model with dictionary extension. The example shows how a German sentence is translated into the English sentence. Different colors are used to represent the aligned phrases. The system found the translations for the underlined OOV words in German. We use solid lines to indict a phrase translated using the segment decoder, and dashed lines to indict dictionary lookup.

## 3 EXPERIMENTS

### 3.1 DATASETS

We use IWSLT 2014 German-English (Cettolo et al., 2014) and IWSLT 2015 English-Vietnamese (Cettolo et al., 2015) as our benchmark datasets. We use 'word' as a basic unit as words are better atoms for phrases, while many other papers use BPE (Sennrich et al., 2015). We think word level experiments fit more into the purpose of examining our model, so in our discussion and analysis, we

**Data:** a source sentence: $x_{1:T}$, a dictionary: $D$, maximum segment length $P$
**Result:** a target sentence: $y$
Compute the representations $\mathbf{s}_{i,j}$ for all phrases given the maximum segment length $P$ ;
$y = $ ;
**while** $y$ *not ends with* $\$\$$ **do**
    Compute phrase-attention score: ;
    $\mathbf{a}_{i,j}^{z_k} = \mathbf{attn}(\mathbf{s}_{i,j}, \mathbf{g}_{\mathbf{z}_{<z_k}})$ ;
    $m, n = \arg\max \mathbf{a}_{i,j}^{z_k}$
    **if** *UNK in* $\mathbf{x}_{m:n}$ **then**
        // Retrieve all translations for $\mathbf{x}_{m:n}$ from $D$;
        // The function `src` retrieves the original text for segment $\mathbf{x}_{m:n}$ ;
        $\mathbf{d}^{z_k} = \texttt{lookup}(D, \texttt{src}(\mathbf{x}_{m:n}))$ ;
        // Select the best candidate ranked by the segment decoder, fixing the attention on $\mathbf{s}_{m:n}$ ;
        $y^{z_k} \leftarrow \underset{d^{z_k} \in \mathbf{d}^{z_k}}{\arg\max} \prod_{t \in z_k} p(d_t^{z_k} \mid \mathbf{s}_{m:n}, \mathbf{g}_{\mathbf{z}_{<z_k}}, \boldsymbol{d}_{<t}^{z_k})$ ;
    **else**
        Generate $y^{z_k}$ following the greedy algorithm (§2.4) ;
    **end**
    Append $y^{z_k}$ to output $y$;
**end**

**Algorithm 1:** NP$^2$MT greedy decoding with dictionary

use the word level models. We run the BPE level experiment for the German-English dataset as a reference.

The IWSLT14 German-English dataset contains approximately 153,000 training pairs, 7,000 development pairs, and 7,000 test pairs from translated TED talks. We apply the same preprocessing as Ott et al. (2018) and remove sentences longer than 175 words.

The IWSLT15 English-Vietnamese dataset contains approximately 133,000 sentence pairs. We follow Luong et al. (2017) and use 1,553 sentence pairs in tst2012 as the development set and 1,268 sentence pairs in tst2013 as the test set.

### 3.2 BASELINES

We compare NP$^2$MT with two baseline models in our experiments:

- Seq2Seq-Att: a sequence to sequence model with an attention module (Bahdanau et al., 2014).

- Transformer: a state-of-the-art machine translation model based on Transformer (Vaswani et al., 2017). We use the model implemented in *fairseq*[2] (Gehring et al., 2017b).

- NPMT: a baseline neural phrase-based machine translation model that only explicitly considers target segments (Huang et al., 2018). We use the original implementation.[3]

Appendix. C contains the implemenation details of the baselines and our model.

### 3.3 RESULTS

The IWSLT 2014 German-English and IWLST 2015 English-Vietnamese test results are shown in Tables 1. NP$^2$MT achieves comparable results to Transformer (Vaswani et al., 2017), and outperforms other baselines in both datasets. NP$^2$MT is not only better but also significantly faster than NPMT (Huang et al., 2018). The proposed dynamic programming (§ 2.3) reduces the complexity from $O(STP)$ in NPMT to $O(TP)$ in NP$^2$MT, where $S$ and $T$ denote the length of source and target

---

[2]https://github.com/pytorch/fairseq
[3]https://github.com/posenhuang/NPMT

sentence respectively, and $P$ is the maximum segment length. Hence it results in approximately 10 times speedup at training time (2 hours vs 24 hours on an Nvidia V100 GPU).

| Model | BLEU | |
|---|---|---|
| | Greedy | Beam |
| BSO | 23.83 | 25.48 |
| Actor-Critic | 27.49 | 28.53 |
| DenseNMT | 29.11 | 30.33 |
| Graph2Seq | 29.06 | 30.66 |
| Seq2Seq with attention | 26.17 | 27.61 |
| NPMT | 28.57 | 29.92 |
| Transformer$_w$ | 31.27 | 32.30 |
| Transformer$_b$ | 33.90 | 34.63 |
| NP$^2$MT$_w$ | 30.99 | 31.70 |
| NP$^2$MT$_b$ | **34.75** | **35.06** |

| Model | BLEU | |
|---|---|---|
| | Greedy | Beam |
| Stanford NMT | - | 23.30 |
| Hard monotonic attention | 23.00 | - |
| DeconvDec | - | 28.47 |
| SACT | - | 29.12 |
| Seq2Seq with attention | 25.50 | 26.10 |
| NPMT | 26.91 | 27.69 |
| Transformer | 29.72 | **30.74** |
| NP$^2$MT | **29.93** | 30.60 |

Table 1: Performance on IWSLT14 German-English (left-side) and IWSLT15 English-Vietnamese (right-side) test set. For BSO (Wiseman & Rush, 2016), Actor-Critic (Bahdanau et al., 2016), DenseNMT (Shen et al., 2018) and Graph2Seq (Xu et al., 2018), Stanford NMT Luong & Manning (2015), Hard monotonic (Raffel et al., 2017), DeconvDec (Lin et al., 2018a) and SACT (Lin et al., 2018b), we take the numbers from their papers. For the Seq2Seq with attention baseline (Bahdanau et al., 2014) and NPMT (Huang et al., 2018), we take the numbers from Huang et al. (2018). We use the subscript '$w$' and '$b$' to denote the word level and BPE level for our model and Transformer.

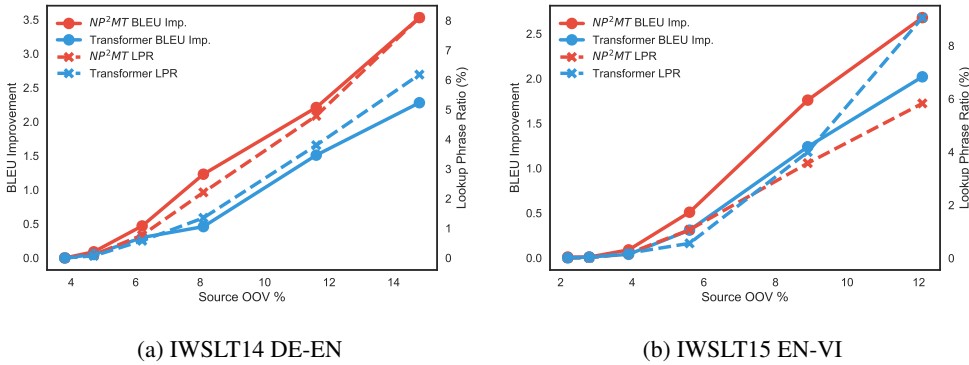

(a) IWSLT14 DE-EN                    (b) IWSLT15 EN-VI

Figure 3: Comparison on BLEU score improvement and lookup phrase ratio under different source language OOV rates both under in-domain dictionaries. NP$^2$MT and Transformer start from BLEU score 30.99 and 31.27 on DE-EN, 29.93 and 29.72 on EN-VI, respectively.

### 3.4 MEMORY-AUGMENTED TRANSLATION

In this subsection, we perform two experiments to understand the effectiveness of augmenting a translation model with a memory module such as a dictionary.

**Rare Words** We first consider a setup where for every word (for Transformer) or every phrase (for NP$^2$MT) that appears in the training data below a certain threshold (e.g., 3 times, 5 times, etc.), we deterministically look up the translation in the dictionary instead of using the segment decoder to generate a target phrase.

We consider six lookup thresholds: 3, 5, 10, 20, 50, 100. For this experiment, we explore two ways to construct the dictionary: using in-domain dataset and out-of-domain dataset (denoted by $D_I$ and $D_O$ respectively). For the in-domain dictionaries, we have 97,399 and 84,817 entries for the IWSLT14

German-English and IWSLT15 English-Vietnamese respectively. For the German-English translation task, we construct $D_O$ from the training set of WMT14 German-English that has 1,243,722 entries.

We show the results in Table 6 and Table 4. Our experiments demonstrate that both $NP^2MT$ and Transformer benefit from the addition of a dictionary, and that the increase in performance is more significant the higher the threshold is. There is no significant difference between using the in-domain dictionary and the out-of-domain dictionary. However, we observe that when the lookup threshold is low, using a bigger dictionary ($D_O$) is consistently better, whereas when the threshold is high $D_I$ is better.

Figure 3 shows the dictionary usage ratio as a percentage of the number of translated words (Transformer) or phrases ($NP^2MT$). Table 3 shows the statistics on the length of the phrases on the IWSLT14 German-English data using threshold 10 model as an example. We can see that the model learns to capture phrases of various length automatically in both the source and the target side.

| | |
|---|---|
| Source | $[\text{dies}]_1$ ist $[\text{die maschine}]_2$ $[\text{unterhalb}]_3$ (von $\underline{\text{genf}}$)$_4$ $[.]_5$ |
| Greedy decoding | $[\text{this is}]_1$ $[\text{the machine}]_2$ $[\text{below}]_3$ $(\text{geneva})_4$ $[.]_5$ |
| Target ground truth | this is the machine below $\underline{\text{geneva}}$ . |
| Source | $[\text{dies}]_1$ ist (eine $\underline{\text{bescheidene}}$)$_2$ $[\text{kleine}]_3$ ($\underline{\text{app}}$)$_4$ $[.]_5$ |
| Greedy decoding | $[\text{this is}]_1$ $(\text{a modest})_2$ $[\text{little}]_3$ $(\text{app})_4$ $[.]_5$ |
| Target ground truth | this is a $\underline{\text{modest}}$ little app . |
| Source | $[\text{unsere}]_1$ ($\underline{\text{zeitschriften}}$)$_2$ $[\text{werden von}]_3$ $[\text{millionen}]_4$ $[\text{gelesen}]_5$ $[.]_6$ |
| Greedy decoding | $[\text{our}]_1$ $(\text{journals})_2$ $[\text{are}]_3$ $[\text{read by}]_5$ $[\text{millions}]_4$ $[.]_6$ |
| Target ground truth | our $\underline{\text{magazines}}$ are read by millions . |

Table 2: DE-EN translation examples, where "$[\cdot]$" denotes the phrase boundary, "$(\cdot)$" indicts a phrase looked up from the external dictionary, and "$\underline{\,\cdot\,}$" represents the frequency of the word replaced by the UNK token in the neural network model. The subscript represents the corresponding phrases alignments discovered by the phrase-attention mechanism (§2.2).

We show translation examples from $NP^2MT$ and lookup threshold 50 in Table 2. $NP^2MT$ discovers meaningful source phrases such as "this is", "the machine" and "read by".

| Length | 1 | 2 | 3 | 4 | 5 | 6 |
|---|---|---|---|---|---|---|
| German | 25.5 | 30.1 | 21.9 | 11.0 | 5.3 | 6.1 |
| English | 65.0 | 24.3 | 7.9 | 2.2 | 0.4 | 0.6 |

Table 3: Phrase length statistics. The ratio (%) of the length of the phrases learned by $NP^2MT$ in the IWSLT14 German-English when we set the threshold as 10.

| Threshold | 3 | 5 | 10 | 20 | 50 | 100 |
|---|---|---|---|---|---|---|
| Transformer | 29.72 | 29.71 | 28.97 | 28.40 | 25.68 | 23.00 |
| Transformer + $D_I$ | 29.72 | 29.72 | 29.01 | **28.71** | 26.92 | 25.02 |
| $NP^2MT$ | 29.93 | 29.78 | 29.27 | 27.87 | 25.89 | 23.63 |
| $NP^2MT+D_I$ | **29.94** | **29.79** | **29.36** | 28.38 | **27.65** | **26.31** |

Table 4: IWSLT15 English-Vietnamese Translation with an in-domain dictionary. We use $D_I$ to denote an in-domain dictionary.

**Cross-Domain Translation** In this experiment, we evaluate our best two models: Transformer and $NP^2MT$ on a cross domain machine translation task with and without memory modules. We train both models on the IWSLT14 German-English dataset (TED talks) and test them on the WMT14 German-English dataset (news articles). The goal of this experiment is to evaluate the effectiveness

| Model | BLEU |
|---|---|
| Transformer | 14.69 |
| Transformer + $D_{\mathrm{WMT}}$ | 15.60 |
| NP$^2$MT | 14.86 |
| NP$^2$MT+$D_{\mathrm{WMT}}$ | **16.11** |

Table 5: Results on cross-domain translation by training the model on IWSLT14 and testing on WMT14, both with German-English data. In the test data from WMT14, OOV rates are 12.8% for German and 6.7% for English.

of the memory module in cases when there are many out of vocabulary words in a more realistic setting. We construct a phrase dictionary using the training set of WMT14, although we do not train our models on this. Future work can explore better setups such as expert-curated dictionaries for cross-domain translation.

We show our results in Table 5. While the addition of a dictionary improves the performance of both models, a phrase-based model such as NP$^2$MT can use this memory module more effectively, as demonstrated by a higher absolute gain (+1.25 vs. 0.91) in terms of BLEU score.

## 4 RELATED WORK

Huang et al. (2018) introduced a variant of neural phrase-based machine translation that only considers target segments. Their model builds upon Sleep-WAke Networks (SWAN), a segmentation-based sequence modeling technique described in Wang et al. (2017a), and uses a new layer to perform (soft) local reordering on input sequences.

Another related work is the segment-to-segment neural transduction model (SSNT) (Yu et al., 2016b;a). Their model assumes monotonic alignment, which is often inappropriate in many language pairs. In contrast, our model relies on a phrasal attention mechanism instead of marginalizing out monotonic alignments using dynamic programming.

There have been other works that propose different ways to incorporate phrases into attention based neural machine translation. Wang et al. (2017b), Tang et al. (2016) and Zhao et al. (2018) incorporate the phrase table as memory in neural machine translation architectures. Hasler et al. (2018) use a user-provided phrase table of terminologies into NMT system by organizing the beam search into multiple stacks corresponding to subsets of satisfied constraints as defined by FSA states. Dahlmann et al. (2017) divide the beams into the word beam and the phrase beam of fixed size. He et al. (2016) uses statistical machine translation (SMT) as features in the NMT model under the log-linear framework. Yang et al. (2018) enhance the self-attention networks to capture useful phrase patterns by imposing learned Gaussian biases. Nguyen & Joty (2018) also incorporates phrase-level attention with transformer by encoding a fixed number of n-grams (e.g. unigram, bigram). However, Nguyen & Joty (2018) only focuses on phrase-level attention on the source side, whereas our model focuses on phrase-to-phrase translation by attending on a phrase at the source side and generating a phrase at the target side. Our model further integrates an external dictionary during decoding which is important in open vocabulary and cross domain translation settings.

## 5 CONCLUSIONS

We proposed Neural Phrase-to-Phrase Machine Translation (NP$^2$MT) that uses a phrase-level attention mechanism to enable phrase-to-phrase level translation in a neural machine translation system. We showed how to to marginalize over all possible segments at training time and presented a greedy algorithm or beam search for decoding. We also proposed to incorporate a memory module derived from an external phrase dictionary to NP$^2$MT to improve decoding. Our experiments showed that NP$^2$MT outperforms the best neural phrase-based translation model (Huang et al., 2018) both in

terms of model performance and speed, and is comparable to a state-of-the-art Transformer-based machine translation system (Vaswani et al., 2017).

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

## A NP²MT BEAM SEARCH ALGORITHM

**Data:** a source sentence: $x_{1:S}$, beam size $B$, maximum segment length $P$, maximum sentence length $N$

**Result:** a target sentence: $y$

Compute the representations $\mathbf{s}_{i,j}$ for all phrases given the maximum segment length $P$ ;

// $\mathcal{Y}_c$ stores completed target sentences, represented as tuples (SEN, PROB). ;

$\mathcal{Y}_c = \{\}$ ;

// $\mathcal{Y}_o$ stores open target sentences, represented as tuples (SEN, SEG, PROB). ;

// The tuple tracks previously generated segments ($\mathbf{g}_{\mathbf{z}_{<z_k}}$), previously generated words in the current

  segment ($\boldsymbol{y}_{<t}^{z_k}$) and the probablity of generating the target sentence so far. ;

$\mathcal{Y}_o = \{(,,1)\}$;

$len = 0$;

**while** *len* $< N$ **do**

    // $\hat{\mathcal{Y}}_o$ stores the open sentences of $len + 1$ ;

    $\hat{\mathcal{Y}}_o = \{\}$ ;

    **for** $m$ *in* $\mathcal{Y}_o$ **do**

        **for** $l = 0$ **to** $P$ **do**

            Compute the distribution of the next output and select the top $B$ words $\mathcal{P}_w$ from it ;

            **for** $w$ *in* $\mathcal{P}_w$ **do**

                $p \leftarrow$ PROB$(m) \times p_w$ ;

                **if** $w$ *not in* $\{\$, \$\$\}$ **then**

                    // $w$ continues the current segment. ;

                    $o_{seg} \leftarrow$ SEG$(m) + w$ ;

                    append $(o_{sen}, o_{seg}, p)$ to $\hat{\mathcal{Y}}_o$ ;

                **else if** $w = \$$ **then**

                    // $w$ ends the current segment.

                    $o_{sen} \leftarrow$ SEN$(m) +$ SEG$(m)$ ;

                    $o_{seg} \leftarrow$ ;

                    append $(o_{sen}, o_{seg}, p)$ to $\hat{\mathcal{Y}}_o$ ;

                **else**

                    // $w = \$\$$ completes a sentence. ;

                    $c \leftarrow$ SEN$(m) +$ SEG$(m)$

                    append $(c, p)$ to $\mathcal{Y}_c$;

                **end**

            **end**

        **end**

    **end**

    Retain only the top $B$ candidates in $\hat{\mathcal{Y}}_o$ ;

    $\mathcal{Y}_o \leftarrow \hat{\mathcal{Y}}_o$;

    $len \leftarrow len + 1$

**end**

**return** $\arg\max(\mathcal{Y}_c)$

**Algorithm 2:** NP²MT Beam Search.

## B  GERMAN-ENGLISH TRANSLATION WITH IN-DOMAIN AND OUT-OF-DOMAIN DICTIONARIES

| Threshold | 3 | 5 | 10 | 20 | 50 | 100 |
|---|---|---|---|---|---|---|
| Transformer | 31.27 | 30.92 | 30.35 | 28.88 | 26.27 | 23.73 |
| Transformer + $D_I$ | 31.27 | 30.97 | 30.65 | 29.34 | 27.78 | 26.01 |
| Transformer + $D_O$ | **31.67** | 31.40 | 31.04 | 29.43 | 27.42 | 25.20 |
| NP$^2$MT | 30.99 | 30.92 | 29.86 | 28.29 | 25.81 | 23.08 |
| NP$^2$MT + $D_I$ | 30.99 | 31.01 | 30.33 | 29.52 | **28.02** | **26.61** |
| NP$^2$MT + $D_O$ | 31.48 | **31.75** | **31.06** | **30.04** | 27.86 | 25.85 |

Table 6: IWSLT14 German-English Translation with in-domain and out-of-domain dictionaries. We use $D_I$ to denote an in-domain dictionary and $D_O$ to denote an out-of-domain dataset.

## C  IMPLEMENTATION DETAILS

**Seq2Seq-Attention**  We use the model architectures and results reported in Huang et al. (2018) as the baselines for Seq2Seq with attention. The Seq2Seq with attention baselines they built consist of a 2-layer BiGRU encoder with 256 hidden units and a 2-layer GRU decoder with 512 hidden units for the IWSLT14 German-English dataset, and a 2-layer BiLSTM encoder with 512 hidden units and a 3-layer LSTM decoder with 512 hidden units [4] for the IWSLT15 English-Vietnamese dataset. Adam algorithm Kingma & Ba (2014) is used for optimization with an initial learning rate of $1e^{-3}$.

**NPMT**  We also use the model architectures and results reported in Huang et al. (2018) for the NPMT baseline. The NPMT models consist of a reordering layer, a 2-layer BiGRU encoder with 256 hidden units and a 2-layer GRU decoder with 512 hidden units for the IWSLT14 German-English dataset, and a reordering layer, a 2-layer BiLSTM encoder with 512 hidden units and a 3-layer LSTM decoder with 512 hidden units for the IWSLT15 English-Vietnamese dataset. The models are optimized using the Adam algorithm with an initial learning rate of $1e^{-3}$.

**Transformer**  We set the number of layers in both encoder and decoder to 6. Our preliminary experiments suggest that increasing the number of layers does not improve performance further.

We use Adam (Kingma & Ba, 2014) as our optimization method and search the best hyperparameters using a validation set. We use a three-stage learning rate scheduler by replacing the exponential decay with a linear one for fast convergence, similar to the RNMT+ learning rate scheduler (Chen et al., 2018; Ott et al., 2018). Specifically, our learning rate is quickly warmed up to the maximum, kept at the maximum value until $50\%$ training iterations, and is then decayed to zero. The maximum learning rate of Transformer is set to $2e^{-4}$.

**NP$^2$MT**  We use 6-layer BiLSTMs to encode at both words and segment level in the source encoder, 6-layer LSTMs as the target encoder, and a 6-layer transformer as the segment decoder.[5] We also use Adam (Kingma & Ba, 2014) as our optimization method. We set the word embedding dimension to 256 In our experiments, we set the maximum learning rate to $1e^{-3}$, weight decay to $1e^{-4}$, and the dropout rate to $0.4$. We set the word embedding dimension to 256. For 6 layer encoder/decoder, NP$^2$MT and Transformer use $\sim$42M and $\sim$75M parameters, respectively.

---

[4]In our preliminary experiments, we reran the NPMT open source code. We found the NPMT doesn't improve with deeper layers.

[5]We also experiment on LSTM segment decoder variant, but the performance is inferior.

