# OpenReview forum: "Neural Phrase-to-Phrase Machine Translation"
_ICLR.cc/2020/Conference — Reject_

### Official Review · AnonReviewer3 · 2019-10-23
**Official Blind Review #3**

**Rating:** 3

**Review:**

This paper presents a phrase-based encoder-decoder model for machine translation. The encoder considers all possible phrases (i.e. word sequences) up to a certain length and compute phrase representations using bidirectional LSTMs from contextual word embeddings computed with another bidirectional LSTM layer. The decoder also considers possible segmentations and computes contextual representations for the previously generated segments. Each word in the current segment is generated by a Transformer model by attending to all phrases in the source sentence. The authors present a dynamic programming method for considering all possible segmentations in decoding. They also present a method for incorporating a phrase-to-phrase dictionary built by Moses into the decoding process.

I like the idea of phrase-to-phrase translation and the relatively simple architecture proposed in the paper. At the moment, however, I am not quite sure how practical their approach is. One reason is the experimental setting. Both of the datasets used in the experiments are quite small and it is not clear how the proposed model performs when several millions of sentence pairs are available for training.

Another reason is that the computational cost of the proposed model is not really clear. The authors state that it is much more efficient than NPMT but it is not clear how it compares to the standard Transformer approach. It seems to me that the computational cost of their model is highly dependent on the value of P (maximum length of phrases).

At first, I thought the decoder was implemented with LSTMs, but I realized that it was actually implemented with a Transformer by reading the appendix. I think this should be explained in the main body of the paper. I am also wondering how the authors’ model compares to a standard seq-to-seq model whose decoder is implemented with a Transformer.

The equation in section 2.2 seems to suggest that the model prefers segmentations with small numbers of segments. I am wondering if there is any negative effect on the translation quality.

Here are some minor comments:

p.2 valid of -> valid
p.4 lookup -> look up?
p.4 forr -> for
p.4 indict -> indicate?
p.5 Table 1 -> Table 1

**Experience Assessment:**

I have published in this field for several years.

**Review Assessment: Checking Correctness Of Derivations And Theory:**

I assessed the sensibility of the derivations and theory.

**Review Assessment: Checking Correctness Of Experiments:**

I assessed the sensibility of the experiments.

**Review Assessment: Thoroughness In Paper Reading:**

I read the paper at least twice and used my best judgement in assessing the paper.

---

### Official Review · AnonReviewer1 · 2019-10-24
**Official Blind Review #1**

**Rating:** 3

**Review:**

This submission belongs to the field of machine translation. In particular, it looks at the problem of phrase-to-phrase translation (previously used state of the art approach) using neural network approaches. The main idea behind this paper is to use a segmental (analogue of phrases) form of neural networks on the source side and attention mechanism to align those segments with segments generated on the target size. This submission additionally describes how an external dictionary can be incorporated using a heuristic approach. I believe this submission could be of wide interest to the machine learning community. I find experimental validation to be satisfactory whilst presentation unsatisfactory for the following reasons:

1) notation

Given that you are dealing with two sets of sequences (source, target side), segments on both sides, attention linking both sides I find it strange that the notation used is not carefully introduced and clearly explained. What is ${\bf g}_{{\bf z}<z_k}$ using precise mathematical language? How it is different from ${\bf g}_{{\bf z}<k}$ and what that is? The same for ${\bf y}_{<t}^{z_k}$, a*, d* and all other variables. In order to help the reader understand your approach it is fundamental to be precise and not ambiguous about each (!) symbol you are using.

2) Algorithm 1, 2 and Figure 1

The algorithmic description was meant to help the reader to understand the process. Unfortunately I have to disagree that is has accomplished this purpose. Please make sure you have unambiguously explained every single term, explicitly say what you are running argmax over, etc. Please also make sure you are discussing/describing the algorithms/figures in your submission. Given the non-trivial nature, lack of proper introduction into the notation used, you cannot simply point the reader to it and not discuss it.

Minor comments:

Please refrain from using "vanilla model" unless you can cite a publication defining exactly what that is.
Please explain how did you derive dictionary.


**Experience Assessment:**

I have read many papers in this area.

**Review Assessment: Checking Correctness Of Derivations And Theory:**

I carefully checked the derivations and theory.

**Review Assessment: Checking Correctness Of Experiments:**

I carefully checked the experiments.

**Review Assessment: Thoroughness In Paper Reading:**

I read the paper thoroughly.

---

### Official Review · AnonReviewer2 · 2019-11-06
**Official Blind Review #2**

**Rating:** 3

**Review:**

This paper proposed an end-to-end phrase-to-phrase NMT model (NP2MT). I think the contribution of this paper is incremental and the idea is of less novelty. In general, the model is largely based on the NPMT model, where modification is the introduce of phrases in the source sentences. Then the author proposed the memory module strategy. In the experiments, the performance improves significant when using out of domain dictionary, but less significant for in-domain dictionary. I also have a concerns about the experiments. The dataset used in this paper seems not convincing to me. By my own experience, the performance on small dataset for either LSTM or Transformer is not stable. The authors just tested the model performance on WMT test set. I think at least the WMT training data should be used for training as well. Another question is the details about the training time and decoding time, since the dynamic programming is used.

**Experience Assessment:**

I have published one or two papers in this area.

**Review Assessment: Checking Correctness Of Derivations And Theory:**

I assessed the sensibility of the derivations and theory.

**Review Assessment: Checking Correctness Of Experiments:**

I assessed the sensibility of the experiments.

**Review Assessment: Thoroughness In Paper Reading:**

I read the paper at least twice and used my best judgement in assessing the paper.

---

### Public Comment · ~Chong_Ruan1 · 2019-10-03
**Is it possible to extend the encoder to Transformers?**

Thanks for your nice work! Now transformers prevail, it is really surprising to see traditional seq2seq models have such impressive performance.
Is it possible to use a Transformer encoder instead? I guess the performance will further improve.

---

### Decision · Program_Chairs · 2019-12-19

**Decision:**

Reject

**Comment:**

This paper describes how they extend a previous phrase-based neural machine translation model to incorporate external dictionaries. The reviewers mention the small scale of the experiments, and the lack of clarity in the writing, and missing discussion on computational complexity. Even though the method seems to have the potential to impact the field, the paper is currently not strong enough for publication. The authors have not engaged in the discussion at all.